# Cystatin SN (CST1) as a Novel Salivary Biomarker of Periodontitis

**DOI:** 10.3390/ijms241813834

**Published:** 2023-09-08

**Authors:** Federica Romano, Francesco Franco, Matteo Corana, Giuliana Abbadessa, Federica Di Scipio, Barbara Pergolizzi, Chiara Castrignano, Mario Aimetti, Giovanni N. Berta

**Affiliations:** 1Department of Surgical Sciences, C.I.R. Dental School, Section of Periodontology, University of Turin, 10126 Turin, Italy; federica.romano@unito.it (F.R.); matteo.corana@gmail.com (M.C.); federica.discipio@unito.it (F.D.S.); mario.aimetti@unito.it (M.A.); 2Department of Clinical and Biological Sciences, University of Turin, 10043 Orbassano, Italy; francesco.franco@unito.it (F.F.); giuliana.abbadessa@unito.it (G.A.); barbara.pergolizzi@unito.it (B.P.); chiara.castrignano@edu.unito.it (C.C.)

**Keywords:** periodontitis, saliva, untargeted proteomics, cystatin SN, periodontal diagnosis, mass spectrometry

## Abstract

Identification of biomarkers could help in assessing periodontal health status and monitoring treatment outcomes. Therefore, the aim of this cross-sectional study was to identify potential innovative salivary biomarkers for the diagnosis of periodontitis using an untargeted proteomic approach. Forty-five healthy non-smoker participants diagnosed as having periodontally healthy conditions (H), severe periodontitis (P), and healthy but reduced periodontium after active periodontal treatment (T) were consecutively enrolled (15 per each group) in the study. A higher number of spots were identified in the proteome of unstimulated whole saliva collected from H and T subjects compared with P group, mainly within the range of 8–40 kDa. Protein spots of interest were analysed by MALDI-TOF-MS, allowing the identification of cystatin SN (CST1) isoform, as confirmed by Western blot. CST1 was markedly expressed in the H group, while it was absent in most P samples (*p* < 0.001). Interestingly, a distinct CST1 expression was observed in saliva from T patients. CST1 was negatively correlated with the percentage of pathological sites (*p* < 0.001) and was effective in discriminating active periodontitis from healthy periodontal status (whether H or T). Therefore, salivary CST1 may be a promising non-invasive biomarker for periodontal disease diagnosis and monitoring.

## 1. Introduction

Periodontitis is a bacterial driven inflammatory disease resulting in the irreversible breakdown of the tooth-supporting tissues [1]. In its more advanced form, it can lead to tooth loss with negative impact on functional and psychological oral health-related quality of life [2]. It can also promote low-grade chronic systemic inflammation contributing to the occurrence of several diseases, notably diabetes mellitus, atherosclerosis, rheumatoid arthritis, and gut diseases [3,4,5]. Considering that severe periodontitis ranks sixth among non-communicable diseases worldwide, with 1.1 billion prevalent cases in 2019, this poses a huge health and socio-economic burden globally [6,7]. Early diagnosis is thus essential to facilitate timely and successful interventions in order to prevent disease complications and to achieve a better long-term prognosis. 

The diagnosis of periodontitis still relies on clinical and radiographic assessments [8,9]. However, this diagnostic system reflects past tissue destruction, and it is limited in detecting disease activity and risk of progression at both patient and site level [10]. Even clinical signs of inflammation, such as bleeding on probing (BoP), are poor predictors of disease progression [11]. There is, therefore, potential benefit in developing additional tools that could help in assessing current and future periodontal disease activity and monitoring treatment outcomes. 

Saliva is a promising biofluid for the identification of biomarkers for advancing the diagnosis of periodontitis [12,13,14]. It can be easily obtained with minimal patient discomfort, and it contains a large variety of locally synthetized or systemically derived molecules, such as DNA, mRNA, protein, metabolites, and ions that could mirror alterations occurring in oral and systemic health status [15]. More than 5000 proteins of both human and bacterial origin have been recently identified [16]. Among human salivary proteins, approximately 73% appear to be unique to saliva, with the remaining 27% overlapping with serum proteins [17]. 

Consequently, proteomics could be a powerful tool for the identification of potential biomarkers for periodontal disease [18,19]. Recently, untargeted procedures have been successfully introduced, with two-dimensional gel electrophoresis (2-DE) coupled to matrix-assisted laser desorption/ionization-time of flight mass spectrometry (MALDI-TOF MS) representing the current state-of-art techniques [20,21]. They allow the high-throughput characterization of proteins and their post-translational modifications providing individual profiles at different body tissue and fluid level. When compared with other matrices, the salivary proteome is particularly attractive because it is easily accessible, and it has revealed a distinct fingerprint for many oral diseases and systemic pathologies with repercussions in the oral cavity [22,23,24]. Nevertheless, not a single marker or combination has yet been validated as a diagnostic test for periodontal diseases and fully translated into clinical practice [12,13,25].

The best option to address this unmet clinical need is to conduct a cross-sectional study. This methodology facilitates an efficient and cost-effective collection of data from a representative cohort of individuals, providing a valuable starting point for future research endeavors.

The aim of this cross-sectional study was to identify potential salivary biomarkers for the diagnosis of periodontitis using an untargeted proteomic approach. Herein, in order to meet such an objective, the salivary protein profile of severe periodontitis was compared with that from treated periodontitis and periodontally healthy status. The null hypothesis was that periodontal status did not have any impact on it.

## 2. Results

### 2.1. Clinical Findings

A population of 45 Caucasian non-smoker individuals participated in this study: 15 periodontally healthy subjects (H group, 8 females/7 males, mean age 43.3 ± 8.8 years), 15 severe untreated periodontitis patients (P group, 10 females/5 males, mean age 48.5 ± 10.1 years), and 15 periodontitis patients with healthy but reduced periodontium after active periodontal treatment (T group, 7 females/8 males, mean age 51.3 ± 8.5 years). Age and gender were similar in the three groups (*p* = 0.066 and *p* = 0.651). Interestingly, the educational level displayed statistically significant differences among the three groups, with 80.0% of H controls having graduated from university compared with 33.3% and 26.7% of patients in the untreated and treated periodontitis groups, respectively (*p* = 0.012). 

Clinical variables are summarized in Table 1. As expected, mean values of all periodontal variables were significantly higher in P group than in H group (*p* ≤ 0.001). Patients who successfully completed the active periodontal therapy presented clinical conditions comparable with those of H subjects, but they had fewer teeth (*p* = 0.002).

### 2.2. Salivary Proteomic Signature

A higher number of protein spots were identified in the salivary samples collected from H and T groups compared to P group, mainly within the range of 8–40 kDa. The spots of interest were excised and analysed by MALDI-TOF-MS in order to identify the corresponding proteins in a fingerprinting mode. Figure 1 is a representative image of the 2-DE salivary proteomic signature of some of the study subjects according to their periodontal status.

The relative scores, expectation, and functional categories of each identified protein are reported in Table 2. Among the identified proteins, a recurrence of the same spot located in the lowest section of the SDS-PAGE (≅17 KDa) was evident. This spot showed a distinct intensity in both H and T group, being far less expressed in P subjects. MALDI-TOF-MS analysis identified it as cystatin SN (CST1). 

### 2.3. Validation of Selected Protein by Western Blot Analysis

Western Blot (WB) analysis, performed to validate the preliminary 2-DE results, allowed the identification of a band of approximately 16.6 kDa aligned with the molecular weight of CST1 protein isoform in both H and T samples. Conversely, the band was blurred or not visible in most of the samples collected from P patients. Figure 2 displays a representative image of the differential expression of CTS1 among the three experimental groups.

Table 3 summarizes the results of WB analysis on the 45 salivary samples examined in the study. CST1 was abundantly expressed in the H group; contrarily, it was absent or faintly present in the majority of the salivary samples from the P group. A re-establishment of CST1 expression was evident in saliva collected from T patients. The differences in the expression level among the groups were statistically significant (*p* < 0.001). 

### 2.4. Correlation Analysis between CST1 Expression and Clinical Variables

As reported in Table 4, salivary CST1 expression was found to be negatively correlated with full-mouth plaque and bleeding indexes and percentage of pathological sites (each *p* < 0.001), but positively correlated with percentage of healthy sites (*p* = 0.01). Conversely, no statistically significant correlation emerged when considering either age or number of residual teeth.

### 2.5. Cluster Analysis

Results of the exploratory cluster analysis are presented in the form of a dendogram in Figure 3. CST1 levels in saliva were able to separate patients with severe periodontitis from both healthy controls (with two exceptions) and treated periodontitis subjects (with one exception) who were clustered together. 

## 3. Discussion

The results of the present research, the first one to analyze concurrently the salivary proteome of periodontally healthy subjects, periodontitis, and treated periodontitis patients, showed that CST1 had significantly different expression levels between periodontal health and disease and was significantly correlated with periodontal clinical parameters. These novel findings should spur future research to determine whether such a biomarker could prove useful as non-invasive diagnostic tool for periodontitis. 

The three groups of enrolled patients were demographically similar in terms of ethnicity, age, and gender, although the T group had a lower number of residual teeth. According to previous studies, age would seem to impact on both salivary flow and protein composition, while gender does not [26,27,28]. Based on the inclusion criteria, the P group significantly differed for the examined periodontal parameters from both H and T subjects, who were found to be comparable except for the different susceptibility to the periodontal disease. 

Over the past decade, a plethora of studies have examined the cellular microenvironment in both healthy and pathological tissues, leading to significant advancements in therapies and interventions designed to manage or prevent various diseases.

As a result, several studies have investigated the effects of CST1 expression on the cellular microenvironment, revealing a correlation between CST1 expression and migratory and invasive behavior of gastric and esophageal cancer cells [29,30]. Similarly, an enhanced expression of CST1 has been observed within peri-implant soft tissues that may influence patients’ susceptibility to peri-implantitis [31]. 

However, a limited number of studies analyzed the activity of salivary CST1 on periodontal tissues.

Salivary CST1 is predominantly produced by the submandibular and sublingual glands and released inside the oral cavity where its main function is the inhibition of human cysteine proteases, namely lysosomal cathepsins B, H, and L [32]. Overall, CST1 plays an important role in the oral microenvironment, primarily by controlling in vivo proteolytic events [33], which are involved in the etiopathogenesis of periodontal disease. 

In the present study, severe periodontitis patients showed a markedly reduced expression of CST1 when compared to periodontally healthy individuals. These findings are consistent with previously reported data from both targeted [34] and untargeted analysis [24,35]. The concentration of CST1 was found to be depleted in periodontitis patients when compared to healthy controls, but still present in sufficient amount to inhibit host-derived cysteine proteases [34]. It was also demonstrated to be significantly higher in healthy controls when compared to both aggressive [24] and chronic periodontitis patients [24,35,36]. In contrast, other studies found salivary cystatins over-expressed in periodontitis patients [37] or failed to detect any relationship between oral health status and their level [38]. 

It is plausible that the lack, or even the reduction, of salivary CST1 may lead to higher levels of protease activity and may contribute to proteolytic destruction of the periodontal tissues, and a subsequent progression of periodontal disease. In this regard, it was suggested that the inflammatory milieu is responsible for the progressive damage to such cytoprotective proteins and for their consequent depletion. It could also be hypothesized that once the inflammatory and bacterial overload has reached a threshold, the host-derived protective mechanisms are reduced. This threshold may vary between subjects and may be responsible for the individual susceptibility in developing periodontal disease [39]. 

The effect of periodontal treatment on salivary level of cystatins has rarely been addressed. One study suggested that it was not affected by non-surgical periodontal treatment [40]. Interestingly, in the present study, successfully treated periodontitis patients showed significantly higher salivary expression of CST1 than untreated patients and comparable with those of healthy controls. When periodontal gold standards (no sites with PPD > 4 mm or PPD = 4 mm with BoP, and FMBS < 10%) have been targeted [41], active periodontal therapy would seem to revert the expression pattern of CST1 in the whole saliva. Consistently, CST1 allowed the differentiation of patients with active periodontitis from both healthy subjects and treated periodontitis patients in the hierarchical cluster analysis. On the contrary, treated patients and periodontally healthy subjects were grouped together. 

The relationship between the expression of CST1 in whole saliva and periodontal status was also confirmed by the correlation analysis. It was negatively correlated with FMPS, FMBS, and the number of periodontal pockets, but positively correlated with the number of healthy sites. In line with the present results, Aboodi et al. [39] observed in an experimental gingivitis model that salivary CST1 increased progressively during the initial inflammatory phase and decreased with its resolution, suggesting the activation of protective pathways to face the increasing bacterial burden on the gingiva and ultimately to re-establish the tissue homeostasis. 

Notably, the strongest negative correlation was observed between CST1 and the number of deep sites (PPD ≥ 6 mm). Lah et al. [42] and Skaleric et al. [43] described lower cystatin concentration at sites with greater PPD. Since residual deep pockets after active periodontal therapy are regarded as risk factors for both disease progression and tooth loss [44], expression of CST1 in saliva could be also a meaningful marker of periodontal stability after treatment. 

Some authors have also attributed immuno-modulatory functions to cystatins, such as modulation of antigen presentation and cytokine production [45]. It was demonstrated that salivary cystatins could stimulate the production of interleukin-6 by human gingival fibroblasts in vitro [46]. Depletion of CST1 in periodontitis patients, resulting in a reduction of their immuno-modulatory properties, could also contribute to the hyper-inflammatory response to pathogenic biofilm. This hyper-inflammatory response of the host has been described as the main pathogenetic mechanism of periodontal tissue destruction [47]. However, it is not possible to establish if the depletion of CST1 in periodontitis patients was a consequence of the proteolytic activity of bacterial enzymes in the oral environment or a lower expression of the protein due to the complex interplay between the host and the pathogens. 

To date, clinical periodontal parameters are of primary importance for the implementation of the diagnostic algorithm [48] and for monitoring the disease status over time. However, it is unclear whether these parameters are sufficient to monitor disease development, ongoing periodontal breakdown, treatment response, and risk for progression, in particular in highly susceptible patients. For this reason, the inclusion of validated biomarkers in the current classification system has been anticipated to provide more accurate staging and grading of periodontitis [1]. Our results suggest that salivary CST1 could be regarded as a useful biomarker not only for the diagnosis periodontitis but also in monitoring the response after active periodontal treatment and during supportive periodontal care. Given the relative simplicity and the non-invasiveness of collecting salivary samples, a biomarker such as CST1 can be assessed several times during periodontal treatment, making it possible to implement a “tailored” therapeutic strategy and verify the effect of the clinical intervention. 

The present results should be confirmed in longitudinal studies and comparing periodontitis patients before and after active periodontal treatment. Moreover, it would be interesting to investigate the expression of CST1 in what is regarded as the precursor of periodontitis, that is gingivitis [49]. If the level of the protein in salivary samples from gingivitis patients will be as high as that in periodontally healthy and successfully treated patients, CST1 should be regarded as a specific biomarker of periodontitis. 

However, this study has some limitations. First, the cross-sectional design prevents us from establishing any conclusion on the causal and temporal relationship between active periodontal disease and the expression of CST1. The second limitation is the lack in saliva samples of housekeeping proteins commonly used in WB analysis as internal loading controls or internal references to normalize the investigated protein expression.

Finally, the relatively limited sample size is also a study limitation, which is the result of the strict inclusion criteria that we set in terms of demographic and lifestyle characteristics to ensure the homogeneity of the experimental groups. 

## 4. Materials and Methods

### 4.1. Study Design and Population

This cross-sectional study was conducted in accordance with the Helsinki Declaration and reported according to the STROBE guidelines. The research protocol was approved by the Institutional Ethics Committee (protocol No. 0050509). Participants were consecutively recruited among patients referred either for periodontal treatment or only for dental check-up to the C.I.R. Dental School, University of Turin (Italy), between January 2021 and October 2022. All eligible patients were informed about the study protocol and provided written informed consent prior to enrolment. For inclusion in the study, subjects had to be systemically healthy, non-smoker or light smoker (<10 cigarettes/day), have a minimum of 16 teeth, and be older than 30 years of age. Exclusion criteria included pregnancy or lactation, alcohol consumption, treatment with drugs known to affect the periodontal status or the periodontal therapy outcomes (i.e., calcium channel blockers, cyclosporine, phenytoin), antibiotic and anti-inflammatory drug intake during the previous 3 months, and hyposalivation/xerostomia interfering with saliva sampling. 

Participants who fulfilled these criteria were recruited into three equal groups based on their periodontal status according to the criteria proposed by the 2017 International World Workshop on the Classification of Periodontal and Peri-implant Diseases and Conditions [1,50,51] as follows:(i)Periodontally Healthy Controls (H) group: no history of periodontitis, no loss of interdental clinical attachment level (CAL), no radiographic evidence of bone loss, and full-mouth bleeding score (FMBS) < 10%;(ii)Severe Periodontitis (P) group: diagnosis of stage III or stage IV generalized periodontitis, CAL loss ≥ 5 mm, probing pocket depth (PPD) ≥ 6 mm, radiographic evidence of bone loss extending to the middle or apical third of the root, ≥30% of teeth at the stage-defining severity level, no periodontal treatment within 12 months before enrolment;(iii)Treated Periodontitis (T) group: reduced but stable periodontium after active periodontal treatment for stage III or stage IV generalized periodontitis, no sites with PPD > 4 mm or PPD = 4 mm with BoP, and FMBS < 10%. Periodontal treatment was conducted according to the EFP S3 level clinical practice guideline for stage I–III periodontitis [41]). When needed, the multidisciplinary management of stage IV periodontitis was implemented according to EFP S3 level clinical practice guideline for stage IV periodontitis [52]. Following the completion of active periodontal therapy, patients were enrolled in a tailored supportive periodontal care program.

### 4.2. Periodontal Examination

A comprehensive periodontal examination that included radiographic assessment was performed by one calibrated examiner. Intra-examiner reliability was assessed by recording duplicate measurements of PPD and CAL from ten non-study patients at two sessions with a 48-hour interval. The intra-examiner variability was 0.14 mm for PPD and 0.16 mm for CAL. The following periodontal biometric parameters were recorded at six sites per tooth, excluding third molars, using a manual periodontal probe (PCPUNC-15, Hu-Friedy, Chicago, IL, USA): presence/absence of plaque, presence/absence of BoP, PPD, gingival recession (Rec), and CAL. Furcation involvement was also recorded at multi-rooted teeth. The FMBS and the full-mouth plaque score (FMPS) were calculated for each subject. 

### 4.3. Salivary Collection and Protein Sample Preparation

The day following the clinical examination to prevent blood contamination, whole unstimulated saliva (5–6 mL) was collected into a graduated polypropylene sterile tube between 8 and 10 am according to the draining method [53]. Participants were asked not to brush their teeth and to refrain from eating and drinking for at least one hour prior to the collection. Each subject was instructed to allow saliva accumulation in the mouth and to drain it into the sterile tube for about 15 min, while sitting on a dental chair.

Total proteins from salivary sol phase (3 mL) were precipitated overnight at −20 °C with 7 mL of glacial acetone:methanol:tributyl phosphate (12:1:1, respectively), and centrifugated at 14,000× *g* for 20 min at 4 °C; the resulting pellet was dissolved in RIPA lysis buffer (150 mM NaCl, 1.0% IGEPAL^®^ CA-630, 0.5% sodium deoxycholate, 0.1% SDS, 50 mM Tris, Sigma-Aldrich, St. Louis, MO, USA) supplemented with protease inhibitor cocktail (Cell Signalling, Danvers, MA, USA), and then centrifuged at 14,000× *g* for 2 min at 4 °C. The supernatant was aliquoted and stored at −80 °C for further examinations. Protein concentrations of the saliva samples were determined by Bradford colorimetric assay (Bio-Rad, Hercules, CA, USA) [54].

### 4.4. Two-Dimensional Gel Electrophoresis

Two-dimensional gel electrophoresis (2-DE) was performed according to the manufacturer’s instructions (GE Healthcare, Milan, Italy), as previously reported [55]. Briefly, salivary samples containing 0.250 mg proteins were mixed with Destreak Rehydration solution (GE Healthcare, Milan, Italy) and 0.5% of IPG buffer (pH 3–10 NL) in a total volume of 140 μL rehydration buffer.

First dimension electrophoresis was performed in Immobiline DryStrips (7 cm, pH 3–10 NL; GE Healthcare, Milan, Italy) and passively rehydrated (20 V, 12 h). Isoelectric focusing with an IPGphor system (GE Healthcare, Milan, Italy) was performed at 200, 500, and 1000 V each for 1 h (step-N-hold), 1000–5000 V (gradient) for 30 min, and 5000 V (step-N-hold) for 3 h. The Immobiline Dry-Strips were then reduced for 30 min in SDS equilibration buffer (50 mM Tris-HCl pH 8.8, 6 M urea, 30% glycerol, 2% SDS) containing 1% (*w*/*v*) DTT, followed by 30 min alkylation in the same equilibration buffer with 2.5% (*w*/*v*) iodoacetamide instead of DTT. 

After the first dimension, SDS-PAGE electrophoresis was run on a gel precast Criterion TGX (Bio-Rad Laboratories, Inc., Hercules, CA, USA), stained with colloidal Coomassie Blue G250 (Bio-Rad, Hercules, CA, USA), and scanned for an evaluation of the differential profile of protein distribution between the three studied groups.

### 4.5. Mass Spectrometry and Protein Identification

Protein spots were excised, transferred in 1.5 mL Eppendorf microtubes, and subjected to trypsin digestion (Proteomics grade, ROCHE, Milan, Italy). Afterward, 2 µL of sample was mixed with an equal volume of α-cyano-4-hydroxycinnamic acid-based matrix (C8982 Sigma Life Science, St. Louis, MO, USA) saturated in 50% acetonitrile. Finally, 0.8 µL aliquots of this mixture were released on the metal target plate of a Microflex^®^ LRF MALDI-TOF mass spectrometer (Bruker Daltonics, Bremen, Germany) in reflector mode. Protein identification was carried out by searching the NCBI and/or Mascot protein database. The parameters used for the search of a protein database with PMF (peptide mass fingerprinting) were as follows: enzyme, trypsin; species, Homo sapiens; pI range, ±1; Mr range, ±20%; missed cleavage sites allowed, 1; minimum peptide hits, 4; mass tolerance, ±100 ppm; modifications, cysteine treated with iodoacetamide to carboxamidomethyl and methionine in the oxidized form.

### 4.6. Western Blot

Total proteins from each saliva sample (30 µg) were separated by SDS-PAGE as previously reported [56]. Gels were transferred to membranes, saturated with blocking solution (5% milk and 0.1% Tween-20 in PBS), and incubated with anti-CST1 primary antibody (1:500; Santa Cruz Biotechnology Inc., Dallas, TX, USA) overnight at 4 °C. Membranes were then rinsed three times and incubated with the appropriate concentration of secondary antibody conjugated with horseradish peroxidase for 1 h at room temperature. Blots were developed with Clarity Western ECL Substrate (Bio-Rad) using ChemiDoc^TM^ Touch Image System (Bio-Rad).

### 4.7. Statistical Analysis

Qualitative and quantitative variables were summarized using absolute frequency and percentage or mean and standard deviation, respectively. Data were first examined for normality by Shapiro–Wilk test, and if they did not achieve normality, analyses were performed using non-parametric methods. Differences between groups were tested using the ANOVA (age, number of teeth) or Friedman test (periodontal variables), followed by pairwise multiple comparisons (Tukey test or Dunn test) for quantitative variables, or Chi-square or Fisher exact test for qualitative variables, as appropriate. The statistical significance of correlations between clinical variables and CST1 expression was determined using the Spearman rank correlation tests. Finally, an exploratory hierarchical cluster analysis using the Ward’s hierarchical method of agglomeration was performed to classify periodontal status based on CST1 expression levels in saliva. Clustering was based on the squared Euclidean distance as a measure of the distance between the observations. Statistical analyses were performed using SPSS (version 28.0, Chicago, IL, USA) and a 5% level of significance was considered for two-tails tests. 

## 5. Conclusions

Within the limitations of the present study, CST1 could be regarded as a potential biomarker of periodontitis. The different expression in untreated periodontitis patients and both treated patients, with a reduced but stable periodontium, and healthy controls, suggest that this salivary protein may be used not only to support the diagnosis but also to monitor treatment response and risk for disease progression. 

Further data are needed to confirm the current findings and to clarify if CST1 plays a role in the etiopathogenesis of periodontal disease and which is the molecular process underlying its expression across different clinical conditions.

## Figures and Tables

**Figure 1 ijms-24-13834-f001:**
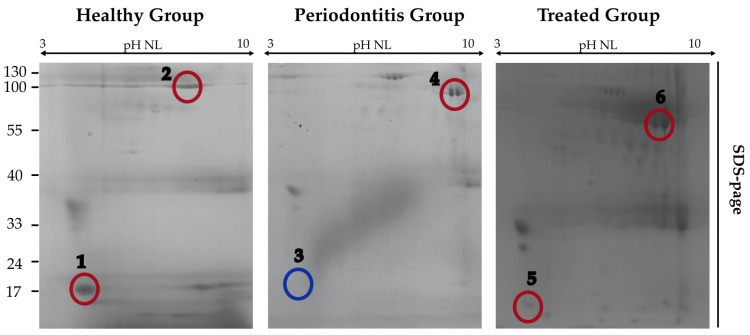
Two-Dimensional Gel Electrophoresis of saliva sol phase samples. Unambiguously identified spots by MALDI-TOF-MS are circled in red and numbered. Spot number 3 is circled in blue, indicating the alleged position of Cystatin SN (CST1) in P samples.

**Figure 2 ijms-24-13834-f002:**
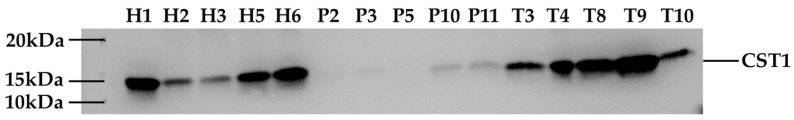
Western blot (WB) analysis of CST1 in salivary samples from 5 healthy subjects (H group), 5 periodontitis patients (P group) and 5 treated subjects (T group).

**Figure 3 ijms-24-13834-f003:**
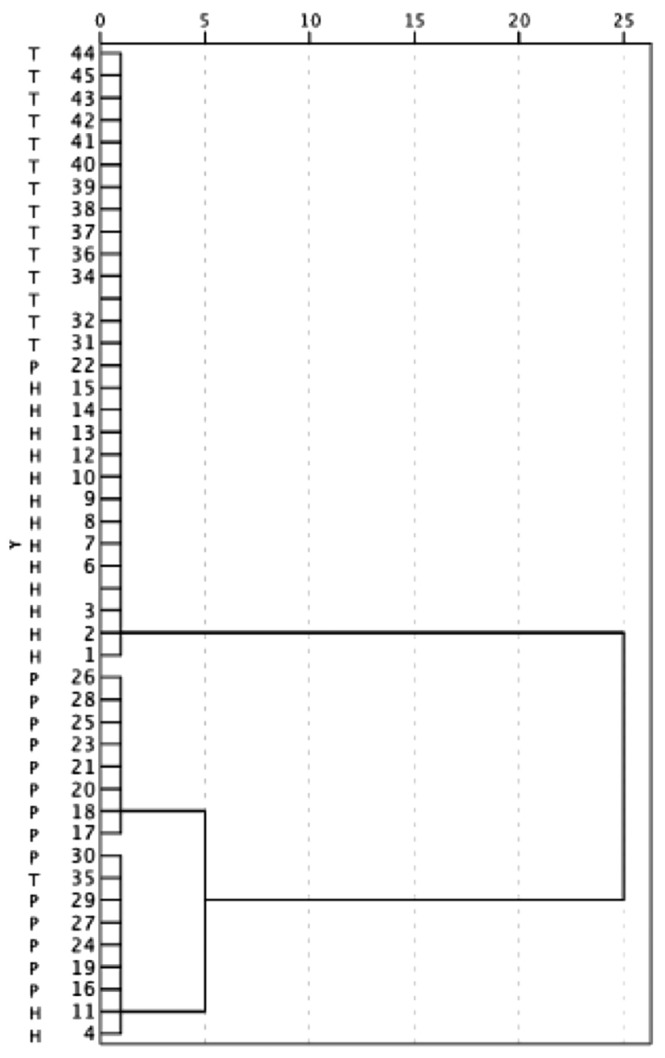
Dendogram obtained from saliva samples: clustering of H, P and T patients.

**Table 1 ijms-24-13834-t001:** Clinical characteristics of the experimental groups (mean ± SD).

Variables	Healthy Controls (H) (n = 15)	Periodontitis (P) (n = 15)	Treated Periodontitis (T) (n = 15)
FMPS (%)	13.4 ± 6.2	79.2 ± 19.3	15.1 ± 10.2
FMBS (%)	8.3 ± 2.7	68.6 ± 19.2	10.8 ± 9.1
% sites with PPD ≤ 3 mm	99.3 ± 0.7	59.8 ± 17.1	97.3 ± 1.7
% sites with PPD 4–5 mm	0.7 ± 0.6	23.8 ± 8.4	2.7 ± 1.8
% sites with PPD ≥ 6 mm	0.0 ± 0.0	16.4 ± 12.7	0.0 ± 0.0
Number of teeth	28.5 ± 1.2	24.1 ± 4.5	23.7 ± 3.4

FMPS: full-mouth plaque score; FMBS: full-mouth bleeding score; PPD: probing depth; SD: standard deviation.

**Table 2 ijms-24-13834-t002:** Identified proteins in the analysed salivary samples.

Functional Category	N Spot	Protein Name	Protein ID	Expectation	MW
Hydrolases/Anti-hydrolases	2	α-amylase1A	P0DUB6	1.3 × 10^−15^	58.4
	4	α-amylase1A	P0DUB6	6.3 × 10^−6^	58.4
	6	α-amylase1A	P0DUB6	2.0 × 10^−15^	58.4
Inflammatory, antimicrobial	1	CST1	P01037	0.63	16.6
	5	CST1	P01037	0.00018	16.6

**Table 3 ijms-24-13834-t003:** Abundance of CST1 expression in salivary samples collected from the three studied groups (+, strongly expressed; +/−, faintly present; −, absent).

Healthy Subjects (H)	CST1 Expression	Periodontitis Subjects (P)	CST1 Expression	Treated Subjects (T)	CST1 Expression
H1	+	P1	+/−	T1	+
H2	+	P2	−	T2	+
H3	+	P3	−	T3	+
H4	+/−	P4	+/−	T4	+
H5	+	P5	−	T5	+/−
H6	+	P6	−	T6	+
H7	+	P7	+	T7	+
H8	+	P8	−	T8	+
H9	+	P9	+/−	T9	+
H10	+	P10	−	T10	+
H11	+/−	P11	−	T11	+
H12	+	P12	+/−	T12	+
H13	+/−	P13	−	T13	+
H14	+	P14	+/−	T14	+
H15	+	P15	+/−	T15	+

**Table 4 ijms-24-13834-t004:** Correlation between CST1 expression in saliva and clinical variables.

Variables	r Value
Age (years)	−0.066
FMPS (%)	−0.657 **
FMBS (%)	−0.628 **
% sites with PPD ≤ 3 mm	0.615 **
% sites with PPD 4–5 mm	−0.596 *
% sites with PPD ≥ 6 mm	−0.811 **
Number of teeth	0.187

FMPS: full-mouth plaque score; FMBS: full-mouth bleeding score; PPD: probing depth; * = *p* value < 0.01, ** = *p*-value < 0.001.

## Data Availability

The datasets used and analyzed in this study are available from the corresponding author upon request.

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
