# Peer review of "Cystatin SN (CST1) as a Novel Salivary Biomarker of Periodontitis"

_ijms, 2023, doi:10.3390/ijms241813834_

Round 1
Reviewer 1 Report
Abstract, line 24: it is speculative to assume that CST1 expression in T group was "re-established", since they were not assessed prior to treatment.
Introduction - line 63: the "state-of-the-art" references are dated 2003, 2010 (20 and 13 years ago, respectively). Please find more recent examples.
Please include in the discussions recent publications studying the link between CST1 and various cancers (gastric, esophageal, laryngeal aso), as well as peri-implant soft tissue (https://pubmed.ncbi.nlm.nih.gov/37060358/
Results - lines 83-86: is university degree a factor in periodontal disease onset/progression?
Since CST1 has been described as a cell-growth promoter in some cases, how does this fit into its role in periodontitis, a disease that is mostly characterised with cell loss and degradation? Please discuss this topic in the discussion/conclusion section.
Author Response
Turin, 31st August 2023
Dear Editor,
Thank you for the comments regarding our manuscript (ID-ijms-2582274) titled “Cystatin SN (CST1) as a novel salivary biomarker of periodontitis” by F. Romano et al
The Reviewers comments spurred us to further improve the overall quality of our message.
Yours faithfully,
Giovanni N Berta
Authors’ responses to Reviewers:
Comments and Suggestions for Authors
Reviewer#1
Comment 1 to the Authors: Abstract, line 24: it is speculative to assume that CST1 expression in T group was "re-established", since they were not assessed prior to treatment.
Authors’ response/action: We thank the referee for the observation, we have made the necessary modifications to the abstract section (line 24).
Comment 2 to the Authors: Introduction - line 63: the "state-of-the-art" references are dated 2003, 2010 (20 and 13 years ago, respectively). Please find more recent examples.
Authors’ response/action: We thank the referee for the suggestion, outdated citations have been replaced with more recent references.
Comment 3 to the Authors: Please include in the discussions recent publications studying the link between CST1 and various cancers (gastric, esophageal, laryngeal aso), as well as peri-implant soft tissue (https://pubmed.ncbi.nlm.nih.gov/37060358/
Authors’ response/action: We express our gratitude for the Reviewer's suggestion, and we have addressed these aspects in the discussion (line210-220)
Comment 4 to the Authors: Results - lines 83-86: is university degree a factor in periodontal disease onset/progression?
Authors’ response/action: Only a limited number of studies have investigated whether the educational background acts as a causal factor rather than merely a correlate of oral diseases. Anyway, higher educational attainment has been associated with a lower risk of periodontitis, but most of the total effect of educational attainment on periodontitis would seem to be mediated through risk factors such as access to dental care and life style behaviors. We made some revisions to the text to prevent any potential bias towards the readers (lines 87-88).
The socio-economic inequalities in oral diseases have been extensively described and are consistent across countries; however, few studies have examined whether such factors are a cause (rather than merely a correlate) of oral health.
Comment 5 to the Authors: Since CST1 has been described as a cell-growth promoter in some cases, how does this fit into its role in periodontitis, a disease that is mostly characterised with cell loss and degradation? Please discuss this topic in the discussion/conclusion section.
Authors’ response/action: We appreciate the reviewer’s observation. We reported in the discussion our speculations about the effect of the observed differential expression of CST1 among the three studied groups (lines 220-222 and lines 233-241)
Reviewer 2 Report
This cross-sectional study aimed to identify potential salivary biomarkers for the diagnosis of periodontitis. The research is interesting but the presentation needs to be improved. The section Material and Methods is between Discussion and Conclusions. I would like to suggest moving it before the section Results as usual. The sample size calculation is missing.
Author Response
Turin, 31st August 2023
Dear Editor,
Thank you for the comments regarding our manuscript (ID-ijms-2582274) titled “Cystatin SN (CST1) as a novel salivary biomarker of periodontitis” by F. Romano et al
The Reviewers comments spurred us to further improve the overall quality of our message.
Yours faithfully,
Giovanni N Berta
Authors’ responses to Reviewers:
Reviewer#2
Comment 1 to the Authors: This cross-sectional study aimed to identify potential salivary biomarkers for the diagnosis of periodontitis. The research is interesting but the presentation needs to be improved.
Authors’ response/action: We appreciate the suggestion provided by the Reviewer, and as a result, we enhanced the presentation of this study (line 71-74).
Comment 2 to the Authors: The section Material and Methods is between Discussion and Conclusions. I would like to suggest moving it before the section Results as usual.
Authors’ response/action: The sections of our manuscript have been structured in line with the template provided by the Journal which places the conclusions at the end of the discussion paragraph.
Comment 3 to the Authors: The sample size calculation is missing.
Authors’ response/action: We appreciate the reviewer’s valuable observation. The analysis we have proposed represents a pilot study conducted on a restricted number of individuals with strict eligibility criteria to examine the expression of CST1 in subjects with comparable background characteristics except for periodontal status. The sample size calculation will be incorporated into a future study, in which we intend to expand the number of participants and to mimic more the clinical setting.
Round 2
Reviewer 2 Report
Congratulations, I think that paper could be published in current form.